# Systematic Review and Meta-Analyses of Aminopeptidases as Prognostic Biomarkers in Amyotrophic Lateral Sclerosis

**DOI:** 10.3390/ijms24087169

**Published:** 2023-04-12

**Authors:** Bárbara Teruel-Peña, José Luís Gómez-Urquiza, Nora Suleiman-Martos, Isabel Prieto, Francisco José García-Cózar, Manuel Ramírez-Sánchez, Carmen Fernández-Martos, Germán Domínguez-Vías

**Affiliations:** 1Department of Health Sciences, University of Jaén, 23071 Jaén, Spain; barbarateruel@correo.ugr.es (B.T.-P.); iprieto@ujaen.es (I.P.); msanchez@ujaen.es (M.R.-S.); 2Department of Physiology, Faculty of Health Sciences, Ceuta University of Granada, 51001 Ceuta, Spain; 3Nursing Department, Faculty of Health Sciences, Ceuta University of Granada, 51001 Ceuta, Spain; jlgurquiza@ugr.es; 4Nursing Department, Faculty of Health Sciences, University of Granada, 18071 Granada, Spain; norasm@ugr.es; 5Department of Biomedicine, Biotechnology and Public Health, University of Cadiz, 11003 Cadiz, Spain; curro.garcia@uca.es; 6Faculty of Pharmacy, University of San Pablo CEU, 28003 Madrid, Spain; carmen.fernandezmartos@ceu.es

**Keywords:** amyotrophic lateral sclerosis, aminopeptidase, biomarker

## Abstract

Amyotrophic lateral sclerosis (ALS) is a fatal neurodegenerative disease characterized by the progressive loss of motor neurons in the spinal cord, brain stem, and cerebral cortex. Biomarkers for ALS are essential for disease detection and to provide information on potential therapeutic targets. Aminopeptidases catalyze the cleavage of amino acids from the amino terminus of protein or substrates such as neuropeptides. Since certain aminopeptidases are known to increase the risk of neurodegeneration, such mechanisms may reveal new targets to determine their association with ALS risk and their interest as a diagnostic biomarker. The authors performed a systematic review and meta-analyses of genome-wide association studies (GWASs) to identify reported aminopeptidases genetic loci associated with the risk of ALS. PubMed, Scopus, CINAHL, ISI Web of Science, ProQuest, LILACS, and Cochrane databases were searched to retrieve eligible studies in English or Spanish, published up to 27 January 2023. A total of 16 studies were included in this systematic review, where a series of aminopeptidases could be related to ALS and could be promising biomarkers (DPP1, DPP2, DPP4, LeuAP, pGluAP, and PSA/NPEPPS). The literature reported the association of single-nucleotide polymorphisms (SNPs: rs10260404 and rs17174381) with the risk of ALS. The genetic variation rs10260404 in the DPP6 gene was identified to be highly associated with ALS susceptibility, but meta-analyses of genotypes in five studies in a matched cohort of different ancestry (1873 cases and 1861 control subjects) showed no ALS risk association. Meta-analyses of eight studies for minor allele frequency (MAF) also found no ALS association for the “C” allele. The systematic review identified aminopeptidases as possible biomarkers. However, the meta-analyses for rs1060404 of DPP6 do not show a risk associated with ALS.

## 1. Introduction

Amyotrophic lateral sclerosis (ALS) is a progressive degenerative motor neuron disease of the brain, spinal cord, and brainstem characterized by weakness in the bulbar and limb muscles with atrophy, spasticity, weight loss, and ultimately instance, respiratory failure, thus being a seriously disabling, incurable and lethal disorder that affects 1 to 3 in 100,000 [1,2,3]. Most ALS patients in an incomplete block state are dependent on invasive ventilation, increasing their survival time to an average of 11 years [4], but that time is considerably reduced from 2 to 5 years after the onset of symptoms if they refuse invasive ventilation [5]. ALS patients can progress from an incomplete blackout state to being completely paralyzed, immobile, and ceasing to communicate verbally while fully conscious and able to control their eyes [6]. Familial ALS (FALS) represents up to 10% of total ALS cases, commonly considered an autosomal dominant inheritable disease, with ~20% of FALS cases being related to mutations in various genes, including Cu/Zn superoxide-dismutase, dynactin 1, alsin, vesicle-associated protein B, senataxin and angiogenin [7,8]. Sporadic ALS (SALS) is less well understood, and it is also more difficult to identify its genetic contributors, but it represents more than 90% of ALS cases as it is probably associated with a multifactorial etiology with an estimated heritability, and it is widely believed to be that genetic factors play a central role [7,9,10]. However, attempts to identify SALS-associated genetic variants using candidate gene approaches are variable and unsatisfactory [9]. Genome-wide association studies (GWAS) help identify common variants that increase susceptibility to disease.

Protein homeostasis (proteostasis), the correct balance between protein production and degradation, is critical to cell physiology. With proteostasis, protein aggregation is prevented, and a correct proteome is maintained during the life cycle of the cell; however, the collapse of proteostasis implies the etiology of neurodegenerative diseases, with ALS being one of the most representative disorders [11]. Thus, ALS may result from an underlying defect in protein catabolism, either by changes in the expression, activity, mutations, or post-translational changes of the proteolytic enzyme. The accumulation of ubiquitin-conjugated proteins in motor neurons of ALS patients appears to be part of an underlying abnormality of intracellular protein metabolism, showing an important role in the etiology of the disease [12]. In an attempt to identify markers that help detect the risk of suffering from ALS, the systematic identification and characterization of aminopeptidases of the central nervous system (CNS) began for the first time. These central aminopeptidases are enzymes responsible for the catabolism of oligopeptides through the hydrolysis of the NH2-terminal residues, including within these enzymes dipeptidyl aminopeptidases and tripeptidyl aminopeptidases.

Different aminopeptidases may participate in the etiology of neurodegenerative diseases. Alzheimer’s disease contributes efficiently to the N-terminal truncation of Aβ [13] and memory impairment associated with changes in the renin-angiotensin system (RAS) modulated by insulin-regulated aminopeptidase (IRAP) [14]. Both aminopeptidase A (APA, EC 3.4.11.7) and dipeptidyl peptidase IV (DPP4 or also known as adenosine deaminase complexing protein 2 or CD26, EC 3.4.14.5) could contribute to the production of pE3-Aβ [13]. Amyloid β1–40/42 can be cleaved by APA, which releases Aβ2–40/42, or by DPP4, which liberates the N-terminal dipeptide, thereby yielding Aβ3–40/42 [13], but in other activities soluble aminopeptidases (alanyl aminopeptidase, MP100 or Puromycin-sensitive aminopeptidase, AlaAP or PSA/NPEPPS, EC 3.4.11.14; arginyl aminopeptidase, ArgAP, EC 3.4.11.6; leucyl aminopeptidase, LeuAP, EC 3.4.11.1; and pyroglutamyl aminopeptidase or Thyrotropin-releasing hormone-degrading enzyme, pGluAP, EC 3.4.19.3) does not suggest or does not allow identification of changes in protein degradation that sensitize neuronal death [15]. Other conditions with neuronal degeneration, such as brain inflammation, show an active role for DPP4, aminopeptidase N (EC 3.4.11.2), as well as dipeptidyl peptidases II (DPP2, E.C. 3.4.14.2), 8 (DPP8, EC 3.4.14.5) and 9 (DPP9, EC 3.4.14.5) and cytosolic AlaAP, being involved in the regulation of autoimmunity and inflammation triggered by ischemia [16]. The search for inhibitors and antagonists of these aminopeptidases shows an important protective role [17,18,19].

Being part of the dipeptidyl dipeptidases family, the activities of dipeptidyl peptidase I, also known as Cathepsin C (DPP1 or CTSC, EC 3.4.14.1), and DPP4 are associated with neuromuscular diseases [20].

Dipeptidyl peptidase VI (DPP6) is located on chromosome 7q36, and its differential expression of the DPP6 gene has been linked to spinal cord injury in rats [21] and is a potential candidate for autism [22]. DPP6, which is also known as VF2, DPPX, DPL1, and MRD33, encodes a dipeptidyl-peptidase-like transmembrane protein (no detectable protease activity due to the absence of the serine residue in the serine protease catalytic domain) expressed predominantly in the brain, with high expression in the amygdala, cingulate cortex, cerebellum, and the parietal lobe. Altered membrane excitability in neuronal populations leads to differential dysfunction and vulnerability specific to neurological diseases, including the SOD1G93A mouse model of ALS with altered expression of TASK1 potassium channels (TWIK-related acid-sensitive K^+^ subunit 1), a member of the KCNK family of two-pore-domain K^+^ channels, whose deficiency contributes to neurodegeneration due to greater excitability and vulnerability of motor neurons [23]. The DPP6 protein (MEROPS: S09.973) The DPP6 protein binds to Kv4-specific neuronal voltage-gated potassium channels (type A or KCND2) to modulate their functional activity [24], alter its expression, biophysical properties, and excitability in the glutamatergic synapse [25], thereby regulating the biological activity of neuropeptides that convert precursors into active forms or with different biological functions. DPP6 establishes the K^+^ current gradient (type A), essential for the proper functioning of the brain, where they act to delay excitation, regulate the frequency of activation in the dendrites of the pyramidal neurons of the hippocampus [26], and promote the growth and stability of filopodia to create synaptic connections as precursors to dendritic spines [27]. DPP6 affects neural and synaptic development, but its deficiency (DPP6-KO) causes impaired learning and memory, as well as a smaller brain [28]. GWAS for SALS suggests different intronic variants of genes that confer SALS susceptibility in various populations, including Single Nucleotide Polymorphisms (SNPs) of DPP6 [29,30]. Collectively, a greater understanding of such underlying disease mechanisms may open the door to new biomarkers and therapeutic interventions for ALS. Biomarkers are essential both for the diagnosis of ALS, as well as for the evaluation of the severity of the pathology, its etiology, and its prognosis. Unfortunately, there is no consensus to have validated biomarkers for measurement in fluids, but the new findings are consistent with low-grade neutrophilia, circulating brain neurofilaments as markers of axoneuronal degeneration [31] and hypoxia as ALS phenotypes [32,33]. Currently, there are validated biomarkers established for clinical use in the detection of ALS from biochemical techniques, metabolic and proteomic studies, imaging, cytological and neurophysiological, being used, among others, lipids that involve the stability of the membrane due to changes in total cholesterol levels [34], together with their lipoprotein fractions, and chitinase proteins as pharmacodynamic and prognostic biomarkers [35].

In an attempt to identify biomarkers with metabolic activity, this work develops a systematic investigation of a type of proteolytic enzymes, aminopeptidases, which comprise the main pathways (cytoplasmic membrane and lysosomal) of CNS and systemic intracellular protein degradation from patients with ALS. The differential regulation of aminopeptidases suggests potential as new biomarkers and possible drug targets for ALS, but multicenter studies are needed to validate these promising therapeutic tools. An evaluation strategy that combines diagnostic mechanisms in pre-symptomatic ALS patients could increase the efficacy of biomarkers.

## 2. Methods

### 2.1. Literature Search and Selection Criteria

Identification of studies was carried out through a multiengine literature search of the PubMed, Scopus, CINAHL, ISI Web of Science, ProQuest, LILACS, and Cochrane databases by 3 independent researchers (B.T.-P., N.S.-M., and G.D.-V.). The review was registered in PROSPERO with ID: CRD42023394858. The following search query was used to retrieve potentially eligible studies from PubMed: ((amyotrophic lateral sclerosis) AND (aminopeptidase OR Dipeptidyl-Peptidases)) and was appropriately modified for the other databases. All potentially eligible studies were retrieved, and their bibliographies were hand searched for additional studies. It only included full-text articles available, written in English or Spanish and published up to 27 January 2023. Given the scant literature on the subject, in addition to being the first time this study has been carried out, no limit of years is applied to the selection of studies. It was considered only GWASs with a case-control or cohort study design, reporting data on the associations between SNPs and the risk of developing ALS. Studies were considered eligible if they evaluated the association between genetic polymorphism and ALS; if they provided effect measures, such as odds ratio (OR) or relative risk, with the 95% confidence interval (CI); if they reported allele and minor allele frequency (MAF) of population groups; if controls were in Hardy–Weinberg equilibrium; and if the study performed replication in an independent dataset. The exclusion criteria were incomplete articles, data published only in abstract form, articles on another subject, review, and meta-analyses articles, studies presenting combined data with other neurological disorders, that were not case-controls, duplicate studies or written in another language that is not part of the inclusion requirements were discarded. Preferred reporting items for systematic reviews and meta-analyses (PRISMA) guidelines were followed for the methodology used in the study [36]. The detailed screening protocol for the studies is depicted in Figure 1.

### 2.2. Quality Assessment and Quantitative Synthesis for Meta-Analyses

The investigators (N.S-M., J.L.G-U., and G.D.-V.) independently extracted data on the first author’s name, location of the study and publication year, study design, ethnicity of the participants, number of cases/controls, SNPs investigated with the candidate gene, GWASs with a measure of association with corresponding 95% CI and *p*-value obtained from the systematic reviews (Table 1). Meta-analyses were performed when at least 2 studies on the same genetic variant and ALS were available, and the genetic variant for ALS risk was reported as significant in at least one systematic review study. The association between ALS and genetic polymorphism was evaluated by calculating pooled ORs and 95% CIs and using the random-effect model to account for variation between the studies. Heterogeneity was evaluated by the χ^2^-based Q statistics and the I^2^ statistics, where *I^2^* with values of 0% does not show heterogeneity and at the opposite extreme with 75%, it presents high heterogeneity; set as I^2^ > 50% to determine the inter-study heterogeneity with statistical significance. The statistical significance of the pooled OR was determined by Z-test. The fixed-effect model was used for studies with no significant heterogeneity. The recessive and dominant genotypes were termed “CC” vs. “CT” and “TT.” For statistical meta-analyses, forest plot graphics were used to represent a stratified analysis based on study quality performed to investigate the potential of such heterogeneous literature findings. Statistical significance in our meta-analyses was set at *p*-value < 0.05. Statistical meta-analyses were performed using StatsDirect Statistical Analysis Software v3.0 (StatsDirect Ltd., Wirral, UK).

The literature reported an association of risk with ALS between different SNPs, but only 8 studies identified the SNP rs10260404 variant of the DPP6 gene as a crucial risk factor. Three random effects (CC and CT, allele C) and one fixed effect (TT) meta-analyses were performed to assess the influence of the genotype in the development of ALS. The number of persons with CC, CT, and TT genotypes in the control and case groups was used. The Egger bias was used for publication bias, the I^2^ for the heterogeneity, and sensitivity analysis to evaluate that none of the studies significantly influenced the results.

## 3. Results and Discussion

### 3.1. Study Selection

After a heterogeneous systematized review of titles and abstracts associating aminopeptidases with ALS, 1626 completed studies were retrieved for further eligibility through search engine databases such as PubMed (n = 27), Scopus (n = 69), CINAHL (n = 20), WOS (n = 92), ProQuest (n = 1138), LILACS (n = 278), Cochrane (n = 2). After excluding 1610 studies for not meeting the inclusion criteria, a total of 16 studies were included in the systematic review, and for the quantitative synthesis with meta-analyses, five studies were used for ALS risk genotype and eight to study the association to ALS by minor allele frequencies (Figure 1).

### 3.2. Study Characteristics

The main characteristics of the studies included in the systematic review include 16 eligible case-control study design about the presence of aminopeptidases as biomolecules present during the etiology of ALS (Table 1). The association of aminopeptidases is collected in 11 studies for SALS, 1 for FALS, and 4 for non-specific ALS. Four studies were conducted in Eastern countries (3 in China and 1 in Russia), and the rest of the studies are distributed in a pooled analysis among Western countries (4 in the United States, 3 in the Netherlands, 2 in Ireland, 2 in Italy, 1 in Belgium, 1 in Canada, 1 in Finland, 1 in Poland, 1 in Switzerland, and 1 in the United Kingdom). Studies of cohorts from China implicate differences in ethnicity. Studies were published from 1983 to 2021. Studies were approved by respective institutional ethics review committees, and informed consent was obtained from all participants. The studies were approved by the institutional ethics review committees of each center, together with the informed consent of all the participants. A total of 11 unique SNPs from the DPP6 gene were reported as significant in all included studies (10 studies with the rs10260404 polymorphism; 1 study for the rs17174381 polymorphism; 1 study for the rs882467 polymorphism; 1 study for the rs11243339, rs3807218, rs830914 polymorphisms, rs2293353, rs2230064, rs3817522, rs1129300, rs3734960).

### 3.3. Quantitative Synthesis

The sample in the control group contributing to a total number of 1873, and in the cases group, it was 1861. I^2^ was 80.1% for the CC meta-analyses (Figure 2A), 61.3% for CT (Figure 2B), and 0% for TT (Figure 2C). The Egger test did not show publication bias in any meta-analyses (*p* > 0.05). 

For the CC genotype, the meta-analytical estimation was OR = 1.18 (95% CI: 0.54–2.59) with *p* > 0.05 (Figure 2A). For the CT genotype, the result was OR = 1.07 (95% CI: 0.83–1.39) with *p* > 0.05 (Figure 2B). For the TT genotype, the result was OR = 0.90 (95% CI: 0.78–1.03) with *p* > 0.05 (Figure 2C).

The sample for the alleles meta-analyses contributed to a total number of 10,678 in the control group and 10,232 in the ALS group. I^2^ was 78.1% in both meta-analyses. When analyzing the OR for allele C, the effect size was 1.13 (95% CI: 0.97–1.30) with *p* > 0.05 (Figure 3A), and for allele T was OR = 0.89 (95% CI: 0.77–1.03) with *p* > 0.05 (Figure 3B). The Egger test did not show publication bias in any meta-analyses (*p* > 0.05). 

The meta-analyses show that the greatest variability in the results for the association of genotype and alleles was found in one of the Chinese cohorts [47] (Table 2 and Table 3).

This review represents the first work that collects a systematic relationship of the different types of aminopeptidases that are involved in the etiology of ALS. There is evidence of the participation of aminopeptidases in neurodegenerative diseases such as Parkinson’s [49] and Alzheimer’s [50], but there is little in the literature that reflects the usefulness of aminopeptidases as biomarkers in ALS. This work represents the first step to collecting aminopeptidases as possible new detection and diagnostic tools for ALS. In summary, this review describes the relationship that the aminopeptidases pGluAP, LeuAP, PSA/NEPPS, and the DPP family (DPP-1, -2, -4, and -6) may have an influence, directly or indirectly, on the severity of ALS and its risk of origin (Figure 4).

Aminopeptidases do not show strong evidence of a generalized alteration in protease activity in the spinal cord in ALS [39]. Only the activity of the cytoplasmic fraction of pGluAP shows significantly altered elevated activity in ALS cases compared to normal controls. Increased pGluAP activity in ALS may represent an adaptation to maintain the excitatory drive of surviving motor neurons through increased processing of TRH to its active metabolite [39]. The lack of evidence for a generalized abnormality in much of the aminopeptidase activities in the spinal cord of ALS cases is likely because it is postmortem tissue resulting from the disappearance of a large population of motor neurons. In contrast, the selective increase in pGluAP activity in ALS suggests some adaptive metabolic response to the disease process, which may be an adaptive response secondary to motor neuron injury and death rather than a primary pathophysiological cause.

Results of an enzyme immunoassay of LeuAP in cerebrospinal fluid and blood serum of ALS patients and other neurodegenerative diseases appear higher just at the onset of symptoms than in the end-stage, but the determination of LeuAP as a biomarker remains unclear associated with the stage of the disease due to the low number of cases and replicates carried out [44]. The absence of detection of LeuAP expression in the biological fluids of end-stage patients may suggest depletion of the reserve in the nervous system due to advanced neurodegeneration and a low population of living motor neurons capable of secreting circulating LeuAP. Detection of LeuAP expression and/or activity in neural tissue lesions and fluids would point to this protease as a possible ALS marker [39,44]. The myelin LeuAP contains antigenic properties contained by different CNS structures and is absent in the biological fluids of healthy people but measurable in the fluids of patients with different forms of neurological pathology [51]. The main limitation of studies with the LeuAP is the low power of adjustment due to the low number of cases; however, it is considered a candidate as a biomarker because its participation is implicated in different neurodegenerative pathologies that manifest molecular alterations similar to ALS [52].

PSA/NPEPPS, considered an enkephalinase [53,54], points to a possible neuroprotective role in neurodegenerative diseases through direct proteolysis of the accumulation of polyglutamine repeats or neurotoxic hyperphosphorylated TAU protein in the spinal cord, brainstem, cortex, hippocampus, and cerebellum of adult and elderly animals, such as occurs with Huntington’s disease [55] and Alzheimer’s disease [37], improving the population number of motor neurons and showing absence of gliosis. PSA/NPEPPS could also play a relevant role in ALS, as it represents a new degradation pathway directed at substrates of neurotoxic proteins of pathological aggregation, including the direct regulation of the cytosolic Cu/Zn superoxide dismutase (SOD1) protein through proteolysis [53]. PSA/NPEPPS is a direct endogenous regulator that controls the abundance and expression of the SOD protein. In addition, the expression of PSA/NPEPPS is significantly decreased in motor neurons of both SOD1G93A transgenic mice and patients with sporadic ALS, suggesting its possible contribution to the pathogenesis of the disease [37,53,56]. Observations suggest that PSA/NPEPPS is a direct endogenous regulator that controls SOD1 protein abundance and expression in response to alterations in intracellular SOD1 levels through a positive feedback mechanism [37]. This is demonstrated with in vitro results in the murine neuroblastoma cell line. The overexpression of the SOD1 protein demonstrates an increase in the expression of the PSA/NPEPPS protein. However, it appears that SOD1 overexpression in vitro may trigger an upregulation of PSA/NPEPPS. However, evidence in vivo from SOD1G93A mice (with the mutated and dysfunctional SOD1 expression found in FALS) and in tissues of the postmortem spinal cord of SALS patients shows dramatically decreased PSA/NPEPPS protein levels [37]. These results suggest that contrary to the short-term effect in vitro, prolonged accumulation of SOD1 mutations downregulates PSA/NPEPPS expression in vivo. Reduced pre-protein expression of PSA/NPEPPS in motor neurons may be a novel contributing factor in ALS pathogenesis, leading to decreased clearance of accumulated SOD1. The importance of this aminopeptidase in other pathologies is also demonstrated. In the cerebral cortex of patients with Alzheimer’s disease, PSA/NPEPPS immunoreactivity for reactive microglia is associated [57].

Other investigations show evidence of abnormalities by identifying in human ALS fibroblasts overexpression of several differentially regulated proteins, including DPP2, which act as upregulated biomarkers in ALS compared to control [40].

In diseases with spinal muscular atrophies, including ALS as a neurogenic muscle disease, they have elevated activities of lysosomal proteases and enzymes that accelerate the biosynthesis of muscle collagen [20,58]. Changes in population numbers of inflammatory fibroblasts and phagocytes occur together with increases in enzyme activities, which are accompanied by a proportionally similar increase in muscle collagen content in ALS [20,59]. The collagenase activities of the DPP1 and DPP4 enzymes present a positive correlation with several of the enzymatic activities related to collagen biosynthesis [20]. Furthermore, these increases in activities are usually positively correlated with the severity of muscle atrophy (rather than with specificity), where most acid hydrolytic and alkaline proteolytic activities are markedly increased only in severely diseased muscles [60]. Despite indicating rapid changes in post-translational modification of muscle collagen in ALS, the amount of collagen accumulated is not always perceived to be always consistent [61], nor are the changes in hydrolytic activities that reflect collagen biosynthesis and processing considered significant in other types of neuropathies with deposition of different types of collagens in the endomysium [59,61]. Other factors that cause variations in the activities are the age of the patient and the evolution of the diseases [61]. The activity levels of lysosomal enzymes in muscle tissue, such as DPP1 and DPP2, are elevated in patients with muscular dystrophies and denervative diseases and are not significant until the last stage of the disease process [60]. DPP1 is significantly elevated in mildly affected dystrophic muscles and other diseased muscles; however, DPP2 is slightly increased in dystrophic and other myopathic muscles but not changed in denervated muscles. It is evident that DPP1 and DPP2 enzymes are actively involved in mediating muscle breakdown in muscular dystrophies and other neuromuscular muscle diseases. Possibly the elevated process of formation and accumulation of collagen in ALS is a consequence of the replacement of degraded muscle tissues, despite having a more relevant role in the pathogenesis by modifying the structure and functionality of the organs involved, as occurs with other tissues [62,63]. Further studies are needed to understand the role of aminopeptidases in coordinating the synthesis and breakdown of muscle cell and connective tissue proteins in ALS and various neuromuscular diseases.

GWAS for SALS implicate a number of candidate genes (DPP6, ITPR2, UNC13A, FGGY, ELP3, KIFAP3, 9p21.2) difficult to replicate in independent populations [21,29,42,64,65,66]. The results highlight the DPP6 gene as a great potential susceptibility candidate for ALS. In the current study, we found in the literature that the strongest association was for the SNP rs10260404, an intronic variant found within the DPP6 gene. However, the meta-analyses for the mutated variant of the DPP6 gene (rs10260404) do not show any evidence that associates it with the risk of manifesting SALS. Genotype and allele analysis also suggest that these SNPs do not contribute to the SALS phenotype. These overall results are similar to results that confirm the non-association of risk in the Asian (Chinese cohorts) and Caucasian (Polish and Italian cohorts) populations, both due to ethnic differences and lack of power to detect common variants [38,42,43,46,47], but paradoxically, it suggests that possession of the associated lower frequency allele (C) within DPP6 increases the risk of SALS development in other cohorts of eastern populations [48] and Western (Irish, Dutch, Belgian, Swedish and American) for SNPs rs882467, rs10260404, and/or rs17174381 [21,29,30,41,42,43,48,64]. These polymorphisms can change or affect splicing, transcription, and translation, leading to abnormal DPP6 proteins, which can cause damage or disrupt or enhance neural network formation and plasticity, synaptic integration and excitation, cytosol calcium signaling, and motor neuron mitochondria, ultimately influencing SALS development [48]. The greatest variability in the association with ALS is found in one of the Chinese cohorts analyzed due to their high ethnic difference (often containing different SNPs and/or false positives) [47], without the statistical significance of the allele frequencies and the frequencies of heterozygotes, homozygotes, and genotypes. Furthermore, contradictory to all the results, another cohort of Han ancestry of mainland Chinese ethnicity shows that carriers with the minor C allele (CC + CT) have a lower risk of developing SALS, being more of a protective factor than a detrimental one [48]. In the Irish and Italian populations [41], the association for rs10260404 was significantly more susceptible to disease for homozygous carriers of the risk allele (homozygous recessive genotype: CC) and minor allele frequency (C) compared with homozygous dominant (TT) and heterozygous (CT) genotypes [21,41]. However, the meta-analyses combined with all the registered studies do not confirm it, despite suggesting a risk associated with the CC genotype and the C allele in a non-significant manner. Other sequence variations in the DPP6 gene correspond to known polymorphisms, identifying six variants that include four benign silent changes without affecting the function of the protein and two missense mutations [43]. Therefore, the role of DPP6 in the pathogenesis of ALS remains unclear. The overall results of this work highlight the difficulty of identifying genetic associations with a rare and genetically heterogeneous disorder such as ALS and suggest that common SNPs are unlikely to represent a substantial proportion of the different ALS subtypes. This is in regard to common variants that increase susceptibility to the disease. On the contrary, rare copy number variants (CNVs) for susceptibility to ALS are little studied. CNVs have been more associated with neuropsychiatric traits, such as autism, schizophrenia, and epilepsy [22,67,68], but CNVs in ALS patients were not powered to detect the association of rare events or do not play an important role in ALS pathogenesis [45,69,70,71].

All of these differences between populations may be due to factors in population size, sample set, and ethnic factors. Our cumulative cohort size is powerful enough to detect a moderate association in motor neuron neurodegeneration, but ethnic specificity is a large variability factor in genetic analysis. Despite the fact that the variations in the DPP6 gene suggest an association with the appearance of SALS, given the heterogeneous origin of the pathology, it is necessary to identify a common polymorphism associated with susceptibility in different countries, and for this, it is necessary to increase the association studies as well as the number of participants to confirm the risk association.

This review has found aminopeptidases involved in the regulation of the renin-angiotensin system that are considered relevant biomarkers in central [72] and peripheral organs [73]. However, their activities are not significant in the neural tissues of postmortem patients with ALS [39]. The notable changes in cell expression of aminopeptidases during Wallerian degeneration of peripheral nerve are known [74]. The common processes that occur in neurodegenerative processes can advantageously turn aminopeptidases into diagnostic tools and therapeutic targets (Figure 4C). Neurofibrillary tangles and neuritic plaques, characteristic of Alzheimer’s disease, are potential substrates for different proteases. The common alterations of neurodegeneration and different factors associated with the metabolic syndrome are present in ALS [75]. Deficiencies of certain types of aminopeptidases cause brain atrophy, loss of neurons in the retina and brain, neurodegeneration of the hippocampus, and impaired learning and memory [76,77,78]. In addition, pharmacological modulation of aminopeptidases could help reduce central hypertension and reduced glutamate excitability [79].

Currently, there are only data on the role of aminopeptidases in postmortem samples and few references at the time the ALS patient is diagnosed, already manifesting the motor syndrome (Figure 4). However, there are no studies that analyze aminopeptidases as early detection tools. It is proposed that aminopeptidases in ALS could be good candidates as prediagnotic secretable-biomarkers and a therapeutic target, as they seem to be recognized in different types of assays for neurodegenerative diseases [57,73,74,80,81,82,83,84], alteration in collagen homeostasis [85], immunoreactivity [16,86,87,88,89,90], neoplasms [91,92,93,94], cirrhosis [95], metabolic syndrome [96], and renal [73,97] and cardiac pathologies [16,82]. Studies are needed during the pre-symptomatic phase to understand the molecular changes that precede the onset of motor symptoms and motor neuron death from hyperexcitability.

## 4. Conclusions

This review aimed to summarize for the first time all the studies that pool data on aminopeptidases available in ALS, both for their measurement identification and to try to find new candidate biomarkers.

The results indicate that variations in aminopeptidase levels may be involved in the course of ALS, being generally less present in the terminal stage of the patient. The candidate aminopeptidases found are DPP1, DPP2, DPP4, DPP6, LeuAP, pGluAP, and PSA/NPEPPS. It is presumed that the genetic variability for DPP6 (SNP rs10260404, located on chromosome 7) is associated with the risk of ALS. However, the combined results of the meta-analyses did not detect a role for the genotype and the susceptibility ‘C’ allele in increasing ALS risk in populations in general, and, more specifically, this association is not demonstrated due to the high variability of ethnic background among Chinese populations. Future large-scale population studies should elucidate genetic associations in these additional populations with different ethnic backgrounds, as well as other non-genetic risk factors for developing ALS (such as environmental factors), as well as avoid including false positives for neuropathies that resemble ALS. The identification of markers could help us better understand the pathogenesis of ALS and could possibly lead to new therapeutic targets with a personalized medicine approach.

## Figures and Tables

**Figure 1 ijms-24-07169-f001:**
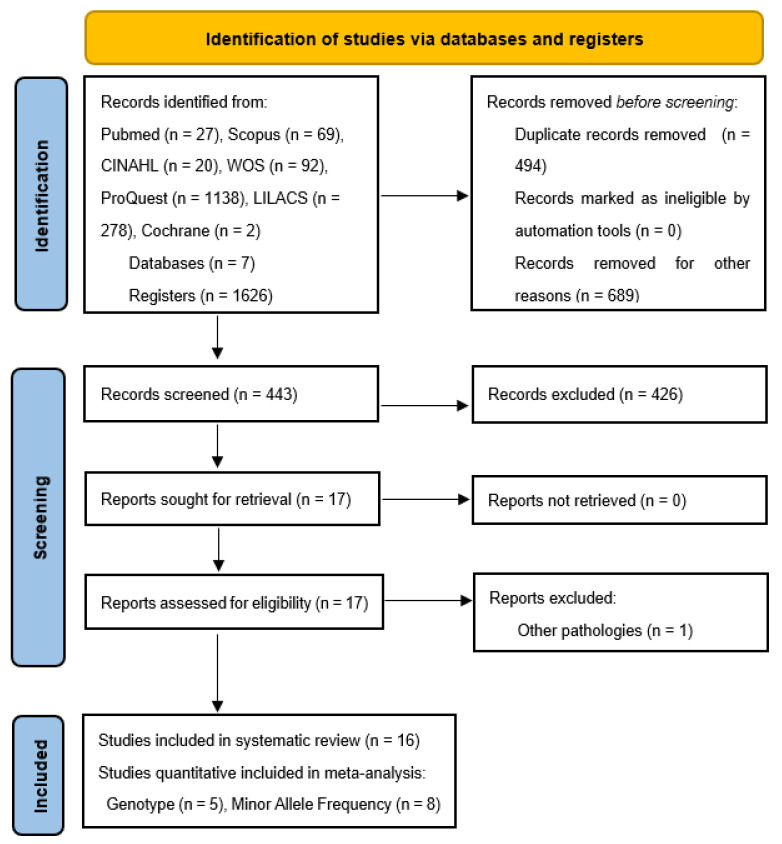
Flow chart of study selection for meta-analyses.

**Figure 2 ijms-24-07169-f002:**
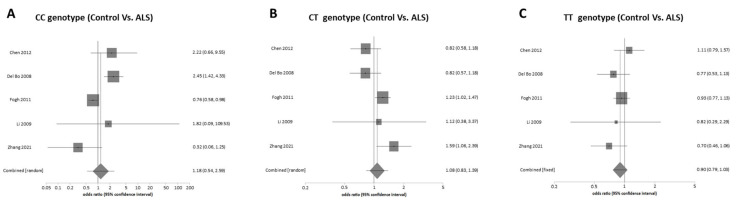
Forest plots reported no significant associations between single-nucleotide polymorphisms (SNP rs10260404) and risk of genotype for sporadic amyotrophic lateral sclerosis (SALS) from meta-analyses. The square size indicates the weight of each study and pooled data, together odds ratio and 95% confidence interval for the difference between control and cases with CC (**A**), CT (**B**), and TT genotype (**C**) between the SALS group and control group [38,41,46,47,48].

**Figure 3 ijms-24-07169-f003:**
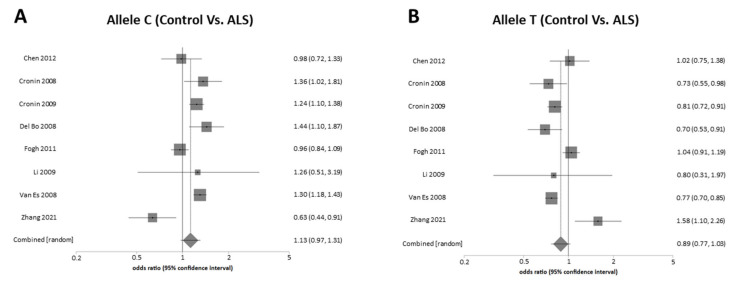
Forest plots reported no significant associations between single-nucleotide polymorphisms (SNP rs10260404) and risk of allele (C > T) of sporadic amyotrophic lateral sclerosis (SALS) from meta-analyses. Despite not being significant, the C allele points to a marked ALS risk (**A**) compared to the T allele (**B**). The square size indicates the weight of each study and pooled data, together with odds ratio and 95% confidence interval Odds ratio meta-analyses for the difference between alleles from control and SALS cases [21,29,30,38,41,46,47,48].

**Figure 4 ijms-24-07169-f004:**
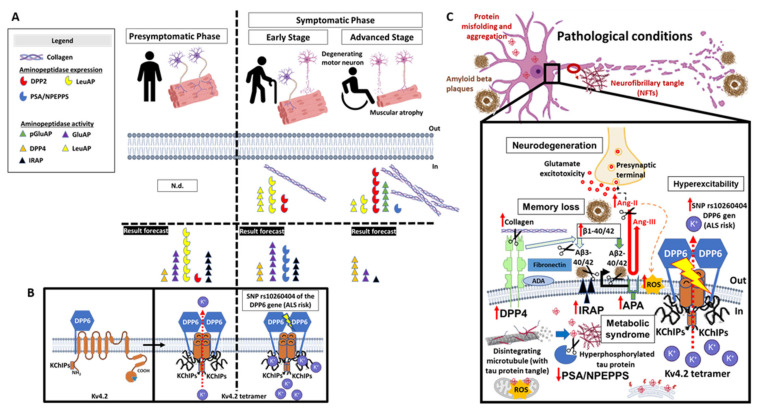
Schematic diagram summarizing (**A**) the most representative results of aminopeptidases found in studies of patients with early and advanced stage amyotrophic lateral sclerosis. The need to use aminopeptidases as pre-diagnosis markers hypothesize the early possibility of detecting the disease before manifesting neurodegenerative motor symptoms (result forecast). The aminopeptidase DPP6 (**B**) binds to voltage-gated potassium channels (Kv4.2), but variations in the expression of the DPP6 gene alter the biological functions of the potassium channel, causing motor neuron hyperexcitability. Schematic representation (**C**) of the deregulation of aminopeptidases that manifest in neurodegenerative diseases (amyotrophic lateral sclerosis, Alzheimer’s disease, Parkinson’s, Huntington’s) with common metabolic processes. The similarity in these processes can become an advantage, and aminopeptidases can be used as a diagnostic tool for the detection of multiple neurodegenerative pathologies. DPP4 is involved in collagen remodeling, but it is also involved in amyloid beta plaque degradation along with IRAP, leading to memory problems (green arrows). The overactivation of the renin-angiotensin system (red curved arrow) contributes to a high production of Ang-II, leading to neurodegeneration due to the release of glutamate and causing ROS by oxidative stress (orange dotted arrow); all of them are factors associated with metabolic syndrome. APA allows the conversion of Ang-II to Ang-III. In ALS patients, the neuroprotective enzyme PSA/NPEPPS is reduced, known to hydrolyze misfolded and aggregated proteins (such as mutated SOD1 and hyperphosphorylated tau) that cause mitochondrial, Golgi apparatus, and nuclear damage. Genetic variants of the DPP6 gene (SNP rs10260404) reduce potassium leakage in Kv4.2 potassium channels (red dotted arrow). ADA: adenosine deaminase; ALS: amyotrophic lateral sclerosis; APA: aminopeptidase A; Ang-II: angiotensin II; Ang-III: angiotensin III; Aβ or β: amyloid-β plaques; DPP2: dipeptidyl peptidase II; DPP4: dipeptidyl peptidase IV; DPP6: dipeptidyl peptidase VI; GluAP: glutamyl aminopeptidase; IRAP: insulin-regulated aminopeptidase activity; LeuAP: leucine aminopeptidase; K^+^: potassium ion; KChIPs: potassium channel-interacting proteins; N.d.: Not determined; NFTs: neurofibrillary tangle; pGluAP: pyroglutamyl-aminopeptidase; PSA/NPEPPS: puromycin-sensitive aminopeptidase; ROS: reactive oxygen species; SNP: single-nucleotide polymorphisms; SOD1: superoxide dismutase 1. The symbol of the scissors represents the hydrolytic activity of aminopeptidase, which cleaves amino acids from the amino terminals (NH_2_).

**Table 1 ijms-24-07169-t001:** Characteristics of the studies included in the systematic review include aminopeptidases as biomolecules for the detection of amyotrophic lateral sclerosis.

Reference	Country (Year)	ALS Type	N	Age	Gender (M:F)	Aminopeptidase	Sample	Result
Ren et al. [37]	United States (2011)	SALS	Control: 6	n.d.	n.d.	PSA/NPEPPS	Postmortem motor neurons (brain and spinal cord tissue)	With decreased PSA/NPEPPS protein expression (*p* = 0.0013), the removal of accumulated SOD1 decreases. PSA/NPEPPS contributes to the pathogenesis of ALS.
			Patient: 19	n.d.	n.d.			
Chen et al. [38]	China (2012)	SALS	Control: 288	n.d	n.d.	DPP6	DNA for SNP gene polymorphism: (rs10260404)	rs10260404 is not associated with the risk of developing ALS in Chinese populations (Genotype distribution, *p* = 0.2; MAF, *p* = 0.9, OR: 1.0, 95% CI: 0.7–1.3).
			Patient: 395	n.d.	n.d.			
Shaw et al. [39]	United Kingdom (1996)	Unspecific ALS	Control: 8	61.0 ± 17.5	6:2	Matrix and cytoplasmic proteases: AlaAP ArgAP DPP3 DPP4 LeuAP GluAP pGluAP Tripeptidyl AP	Postmortem spinal cord	No evidence of generalized alterations in protein-catabolizing enzyme activities in spinal cord tissue was found in ALS. Only pGluAP showed significantly altered activity (119%) in ALS cases compared to normal controls (Control: 2.6 ± 0.6 vs. ALS: 5.7 ± 1.1; *p* < 0.5).
			Patient: 10	67.7 ± 15.4	7:3			
			Control: 8	58.3 ± 12.7	7:1	Lysosomal proteases: DPP1 DPP2		
			Patient: 9	67.0 ± 7.9	4:5			
Narayan et al. [40]	United States (2016)	Unspecific ALS	Control: 5	60.6 ± 7.3	2:3	DPP2	Cultures of human fibroblasts	The differential regulation of DPP2 protein expression in ALS fibroblasts has the potential as a biomarker and potential drug target for ALS. The expression of DPP2 is overexpressed upwards (×2.2) with respect to the controls (*p* = 7.75 × 10^−9^).
			Patient: 4	60.8 ± 6.3	3:1			
Takala et al. [20]	Finland (1983)	Unspecific ALS	Control: 9	43 (25–58)	n.d.	Lysosomal hydrolase: DPP1 Non-lysosomal hydrolase: DPP4	Muscle biopsy and serum samples for diagnostic purposes.	The DPP1 and DPP4 activities in patients with ALS and the control group are not altered. DPP1 (*p* < 0.05; r = 0.35) and DPP4 (*p* < 0.01; r = 0.59) activities are positively correlated with collagen-forming activity (muscle galactosylhydroxylysyl glucosyltransferase; M-GGT). Also, DPP1 (*p* < 0.05; r = 0.44) and DPP4 (*p* < 0.05; r = 0.43) activities correlate with the serum concentration of N-terminal propeptide of type III procollagen (S-PRO III). DPP1 (*p* = n.s.; r = 0.15) activity does not have a significant correlation with muscle prolyl hydroxylase (M-PH) activity, except for DPP4 (*p* < 0.05; r = 0.38) activity, which presents a positive correlation. DPP1 (*p* = n.s.; r = 0.11) and DPP4 (*p* = n.s.; r = 0.16) activities do not correlate with muscle collagen (muscular hydroxyproline, M-HYP). DPP1 activity does not correlate with the degree of severity of muscle atrophy (*p* = n.s.; r = 0.12), but DPP4 activity does (*p* < 0.01; r = 0.53).
			Patient: 8	52 (23–74)	n.d.			
Cronin et al. ^§^ [29]	Ireland, the United States, Netherlands (2008)	SALS	Control: 932	n.d.	n.d.	DPP6	DNA for SNP gene polymorphism: (rs10260404)	Combined GWA analysis suggests that possession of the associated allele within the DPP6 gene increases the risk of ALS (*p* = 2.53 × 10^−6^; OR = 1.37; 95% CI = 1.20–1.56). The strongest association is for the variant in the gene encoding DPP6 and for SNP rs10260404, an intronic variant found within the DPP6 gene on chromosome 7.
			Patient: 958	n.d.	n.d.			
Van Es et al. ^§^ [21]	United States, Netherlands, Sweden, Belgium (2008)	SALS	Control: 1916	n.d.	n.d.	DPP6	DNA for SNP gene polymorphism: (rs10260404)	The SNP rs10260404 polymorphism of the DPP6 gene is strongly associated with ALS susceptibility in different populations of European descent (*p* = 5.04 × 10^−8^; OR = 1.30; 95% CI = 1.18–1.43).
			Patient: 1767	n.d.	n.d.			
Del Bo et al. ^§^ [41]	Italy (2008)	SALS	Control: 239	n.d. (matched for age)	n.d.	DPP6	DNA for SNP gene polymorphism: (rs10260404)	The genetic variant of DPP6 (SNP rs 10260404) is associated as a possible risk factor for developing sALS in an Italian population (Genotype, *p* = 0.0027; MAF, *p* = 0.0066). The CC genotype and C allele are associated with an increased risk of sALS in recessive and allelic association tests (CC vs. CT/TT: *p* = 0.0008; OR = 2.44; 95% CI = 1.41–4.32; C allele vs. to T allele: *p* = 0.0055; OR = 1.43; 95% CI = 1.10–1.86).
			Patient: 266	58.4 ± 12.9	173:93			
Kwee et al. [42]	United States (2012)	SALS	Control: 961	63.5 ± 11.5	755:206	DPP6	DNA for 2 SNP gene polymorphisms: (rs17174381) (rs10260404)	Through the GWA, the SNP rs17174381 (*p* = 4.4 × 10^−4^; OR = 1.9; 95% CI = 1.3–2.8) located in the DPP6 gene shows evidence of association with ALS, but no association with the SNP rs10260404 (*p* = 0.97; OR = 1.0; 95% CI = 0.9–1.1) of the same gene.
			Patient: 183	57.4 ± 12.1	115:68			
Daoud et al. [43]	Canada (2010)	FALS SALS	Control: 190	n.d.	n.d.	DPP6	Peripheral blood DNA for 8 SNP gene polymorphisms: (rs11243339) (rs3807218) (rs56091483) (rs2293353) (rs2230064) (rs3817522) (rs1129300) (rs3734960)	Mutations in the DPP6 genes do not show evidence of being the cause of ALS. The MAF in the ALS cohort did not differ from the dbSNP. rs11243339 (*p* = 0.48) rs3807218 (*p* = 0.14) rs56091483 (n.d.) rs2293353 (*p* = 0.27) rs2230064 (*p* = 0.35) rs3817522 (*p* = 0.32) rs1129300 (*p* = 0.49) rs3734960 (*p* = 0.08)
			Patient: 190 (FALS: 110) (SALS): 80	n.d.	n.d.			
Khokhlov et al. [44]	Russia (1992)	Unspecific ALS	Control: 14	n.d.	n.d.	LeuAP	Cerebrospinal fluid	In patients with ALS, by enzyme immunoassay in the cerebrospinal fluid, the level of the LeuAP enzyme was significantly higher at the onset of the disease with moderate clinical manifestations than in the cases of generalized form and in the terminal phase.
			Patient: 2	57.5 ± 2.1	n.d.		Blood serum	The LeuAP enzyme was not registered in the blood serum of control donors, but the frequency of registration in ALS compared to other neurological patients is approximately 50%.
Blauw et al. [45]	Netherlands (2010)	SALS	Control: 14618	n.d.	n.d.	DPP6	DNA for SNP gene polymorphism: (rs10260404)	The SNP rs10260404 of the DPP6 gene shows ALS susceptibility (*p* = 1.4 × 10^−3^; OR = 2.64) in ALS association analysis but does not explain the associations with the number of rare copy number variables (CNV; Average all CNVs per individual, *p* = 0.28). There is no evidence of a higher global burden of CNV in ALS cases than in controls or a difference in gene content in ALS cases for CNVs in general or for large CNVs (>500 Kb; Average CNVs per individual, *p* = 0.99). The results for the DPP6 locus do not appear to be population-specific (Woolf test, *p* = 0.60). Although the association of the SNP rs10260404 in the DPP6 gene with susceptibility to ALS has been described, the signal from the CNV locus is not explained by rs10260404. CNVs in patients with ALS do not have statistical power to detect the association of rare events.
			Patient: 4434	n.d.	n.d.			
Fogh et al. ^§^ [46]	Italy (2011)	SALS	Control: 1036	n.d.	n.d.	DPP6	DNA for SNP gene polymorphism: (rs10260404)	The MAF for the risk C allele was 0.38 in ALS cases and 0.39 in controls (*p* = 0.64; OR = 0.97; 95% CI = 0.84–1.09). The results of the GWA study do not suggest an association of the DPP6 gene with the risk of ALS susceptibility (*p* = 0.64).
			Patient: 904	n.d.	n.d.			
Cronin et al. ^§^ [30]	Ireland, Netherlands, Poland, United States (2009)	SALS	Control: 1336	60 ± 4.7	706:630	DPP6	DNA for SNP gene polymorphism: (rs10260404)	A pooled analysis of GWA data shows a stronger clustered allelic association at rs10260404 in the DPP6 gene (*p* = 2.62 × 10^−4^; OR = 1.23; 95% CI = 1.1–1.38), showing the same risk allele (C) of the same SNP in each of the populations.
			Patient: 1267	58 ± 2.4	720:547			
Li et al. ^§^ [47]	China (2009)	SALS	Control: 52	45.3 ± 15.1	28:24	DPP6	Genomic DNA extracted from leukocytes of a whole blood sample. DNA for SNP gene polymorphism: (rs10260404)	GWA studies report several SNP polymorphisms that are susceptible to ALS in the Chinese population but do not find significant differences of the SNP rs10260404 of the DPP6 gene between the Chinese ALS group and the control group genotype frequencies (*p* = 0.68; OR = 1.21; 95% CI = 0.48–3.06) and allele frequencies (*p* = 0.59; OR = 1.26; 95% CI = 0.55–2.87).
			Patient: 58	46.4 ± 14.0	34:24			
Zhang et al. [48]	China (2021)	SALS	Control: 261	65.7 (65–75)	148:113	DPP6	DNA for 2 SNP gene polymorphisms: (rs882467) (rs10260404)	SNP rs10260404 of the DPP6 gene is strongly associated with sALS in subjects of Chinese descent and plays roles in ALS pathogenesis, affecting splicing, transcription, and translation of the DPP6 gene (genotype, *p* = 2.84 × 10^−2^). C minor allele of rs10260404 shows a lower risk of developing sALS compared with subjects of other genotypes (*p* = 0.009; OR = 0.64; 95% CI = 0.45–0.90). The minor allele (C) of rs10260404 represents a protective genetic factor.
			Patient: 239	47 (45–65)	143:96			

^§^ Data used for meta-analyses. (M:F): Male/Female; dbSNP: Single Nucleotide Polymorphism Database; FALS: familiar amyotrophic lateral sclerosis; GWA: genome-wide association; MAF: minor allele frequency; n.d.: non data; n.s.: not significant; SALS: sporadic amyotrophic lateral sclerosis; SNP: Single Nucleotide Polymorphism.

**Table 2 ijms-24-07169-t002:** Characteristics of the studies included in the meta-analyses. Genotype frequencies in variant SNP rs10260404 within DPP6 gene between SALS group and control group.

Genotype Frequencies (n (%))						
	Homozygous Recessive		Heterozygous		Homozygous Dominant	
Author	CC Control	CC sALS	CT Control	CT sALS	TT Control	TT sALS
Chen et al., 2012 [38]	4 (1.39)	12 (3.04)	84 (29.17)	100 (25.32)	200 (69.44)	283 (71.64)
Del Bo et al., 2008 [41]	23 (9.6)	55 (20.6)	118 (49.4)	118 (44.4)	98 (41)	93 (35)
Fogh et al., 2011 [46]	169 (16.31)	116 (12.83)	462 (44.6)	449 (49.67)	405 (39.09)	339 (37.5)
Li et al., 2009 [47]	1 (1.92)	2 (3.45)	9 (17.31)	11 (18.98)	42 (80.77)	45 (77.59)
Zhang et al., 2021 [48]	10 (3.88)	3 (1.26)	168 (65.11)	178 (74.79)	80 (31.01)	57 (23.95)

**Table 3 ijms-24-07169-t003:** Characteristics of the studies included in the meta-analyses. Risk of minor allele frequencies (C allele) and *p*-values for studies of variant SNP rs10260404 within the DPP6 gene between the SALS group and control group.

Allele frequencies			Minor Allele Frequency (MAF): Risk Frequency of the C > T Allele					
Author	Control (n (%))	ALS (n (%))	Control MAF		sALS MAF		*p*-Value	OR (95% CI)
			C allele Frequency (n (%))	T allele Frequency (n (%))	C Allele Frequency (n (%))	T allele Frequency (n (%))		
Chen et al., 2012 [38]	576 (100)	790 (100)	92 (15.97)	484 (84.03)	124 (15.70)	666 (84.3)	0.89	0.98 (0.73–1.31)
Cronin et al., 2008 [29]	422 (100)	442 (100)	145 (34.4)	277 (64.6)	184 (41.6)	258 (58.4)	0.03	1.36 (1.03–1.79)
Van Es et al., 2008 [21]	3832 (100)	3534 (100)	1357 (35.4)	2475 (64.6)	1470 (41.6)	2064 (58.4)	5.04 × 10^−8^	1.30 (1.18–1.43)
Del Bo et al., 2008 [41]	478 (100)	532 (100)	164 (34.3)	314 (65.7)	228 (42.9)	304 (57.1)	6.60 × 10^−9^	1.43 (1.11–1.85)
Fogh et al., 2011 [46]	2072 (100)	1808 (100)	808 (39)	1264 (61)	687 (38)	1121 (62)	0.64	0.97 (084–1.09)
Cronin et al., 2009 [30]	2672 (100)	2534 (100)	962 (36)	1710 (64)	1039 (41)	1495 (59)	2.62 × 10^−4^	1.23 (1.1–1.38)
Li et al., 2009 [47]	104 (100)	116 (100)	11 (10.58)	93 (89.43)	15 (12.94)	101 (87.06)	0.59	1.26 (0.55–2.87)
Zhang et al., 2021 [48]	516 (100)	476 (100)	100 (19.4)	416 (80.6)	63 (13.2)	413 (86.8)	0.01	0.64 (0.45–0.90)

## Data Availability

Not applicable.

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
