# Peer review of "Systematic Review and Meta-Analyses of Aminopeptidases as Prognostic Biomarkers in Amyotrophic Lateral Sclerosis"

_ijms, 2023, doi:10.3390/ijms24087169_

Round 1

Reviewer 1 Report

The review on Systematic Review and Meta-analyses of Aminopeptidases as Prognostic Biomarkers in Amyotrophic Lateral Sclerosis is well planned and organized  The introduction is clear. The reference collection is appropriate and the meta data analysis is clear and done very well.  The analysis of results are clear. The discussions needs modification and explain point wise with the clear data analysis. In conclusion provide a hypothesis with clear proposal on the role of Aminopeptidases as Prognostic Biomarkers as a figure and detailed write up. The paper needs English editing.

Author Response

I thank the reviewer for his contributions to improve the manuscript. The discussion has been rearranged to explain in more detail each of the aminopeptidases analyzed and the importance of each as biomarkers has been detailed. The changes are made in yellow.

A new figure has also been added that allows us to simplify the results, the hypothesis of expected results in future studies, and the molecular mechanisms involved in the alterations of aminopeptidases in the degenerated motor neuron.

Reviewer 2 Report

The authors extensively analyzed the literature and performed a Meta-analyses on the Aminopeptidases as Prognostic Biomarkers in Amyotrophic Lateral Sclerosis. They did not find a gene or a polymorphism strictly related to the onset or progression of ALS, having obtained data on the absence of risk association between the SNP rs10260404 in the DP6 gene with the ALS susceptibility.

The manuscript is well written, and the analyzed papers were clearly described and analyzed. Thus, in my opinion the manuscript can be accepted in the present form for publishing in IJMS journal.

Author Response

I thank the reviewer for his time and comments. At the request of other reviewers, to further improve the understanding of the study, a new figure has been added that allows summarizing the details analyzed and explaining the molecular mechanisms involved in the regulation of aminopeptidases in neuron degeneration, as well as the role of aminopeptidases as biomarkers and therapeutic targets.

Thank you.

Reviewer 3 Report

The article is quite well written. Only the introduction need to be corrected a bit, since it is a bit overloaded. Some information should be moved to discussion. 

It should also be mentioned that some of the studies in the analysis have low statistical power (eg. 14 controls against 2 cases). It is hard to get conclusions from such studies. 

Finally, it should be more emphasis should be placed on markers that are already used in clinical investigation and how much really aminopeptidases can contribute to the field.

Author Response

I thank the reviewer for his time and comments for improving this manuscript. The introduction has been reduced and material that would actually explain more of the review has been moved to discussion, emphasizing each aminopeptidase as a biomarker more consistently. The limitation of the scarcity of studies for the use of LeuAP as a biomarker is mentioned in the manuscript, and it is supported by the use of this biomarker in other neurodegenerative diseases that share molecular alterations, as well as its use in other pathologies.

All modifications appear in yellow. A new figure has also been added (Figure 4) that allows us to simplify the results, the hypothesis of expected results in future studies, and the molecular mechanisms involved in the alterations of aminopeptidases in the degenerated motor neuron. In addition, it is specified that in addition to being a biomarker, its pharmacological modulation can also be used as a therapeutic target.